# Salivary Complaints in Burning Mouth Syndrome: A Cross Sectional Study on 500 Patients

**DOI:** 10.3390/jcm12175561

**Published:** 2023-08-26

**Authors:** Federica Canfora, Elena Calabria, Gianrico Spagnuolo, Noemi Coppola, Niccolò Giuseppe Armogida, Cristina Mazzaccara, Domenico Solari, Luca D’Aniello, Massimo Aria, Giuseppe Pecoraro, Michele Davide Mignogna, Stefania Leuci, Daniela Adamo

**Affiliations:** 1Department of Neuroscience, Reproductive Sciences and Dentistry, University of Naples Federico II, 80131 Naples, Italy; federica.canfora@unina.it (F.C.); daniela.adamo@unina.it (D.A.); 2Department of Health Sciences, School of Dentistry, University Magna Graecia of Catanzaro, 88100 Catanzaro, Italy; 3Therapeutic Dentistry Department, Institute for Dentistry, Sechenov University, 119991 Moscow, Russia; 4Department of Molecular Medicine and Medical Biotechnology, University of Naples Federico II, 80131 Naples, Italy; 5CEINGE Advanced Biotechnologies, 80145 Naples, Italy; 6Department of Social Sciences, University Federico II of Naples, 80138 Naples, Italy; 7Department of Economics and Statistics, University Federico II of Naples, 80138 Naples, Italy

**Keywords:** burning mouth syndrome, saliva, xerostomia, sialorrhea

## Abstract

Background: Xerostomia and sialorrhea often accompany Burning Mouth Syndrome (BMS) despite no change in saliva quantity. This study analyzed BMS patients with different symptom combinations: burning only (B), burning and xerostomia (BX), burning and sialorrhea (BS), and burning with xerostomia and sialorrhea (BXS), using a large sample of 500 patients from the University of Naples Federico II. Methods: After a medical evaluation, patients were divided into four groups based on their reported symptoms: B (140), BX (253), BS (49), and BXS (58). Patient data on education, BMI, smoking/alcohol habits, comorbidities, medication use, pain intensity, quality, and psychological profile were collected. Results: The BX group showed a higher prevalence of patients taking blood thinners. Additional symptoms varied among groups, with the BX group experiencing more dysgeusia and globus, and the BS group reporting more tingling. Multivariate analysis identified BMI, dysgeusia, globus, and blood thinner use as significant factors in the B and BX groups, while male gender, tingling, alcohol use, and pain quality were significant in the BS and BXS groups. Conclusions: Overall, BMS patients display a complex range of symptoms, with xerostomia being the most frequent additional symptom. Sociodemographic, psychological, and medical factors cannot fully explain the variations in symptomatology among different patient subgroups. Further research is needed to understand the underlying causes and develop tailored treatment approaches.

## 1. Introduction

Saliva plays a crucial role in maintaining oral health and function by protecting intraoral structures, facilitating digestion and articulation of speech [1]. It helps to maintain a normal pH range, moisturizes the mucous membranes and prevents dehydration [2]. The significance of saliva extends beyond oral health, as it is essential for clear and coherent speech, particularly for public speakers [1].

Saliva production is a complex process involving the three pairs of major salivary glands, (parotid, submandibular and sublingual) which contribute to 90% of saliva secretion, with the remaining 10% derived from minor salivary glands in the oral cavity [1,3].

Under normal conditions the average daily saliva production ranges from 0.5 to 1.5 L with individual variations. However, both psychological state and functional activities, like eating or sleeping, can influence both the quality and quantity of saliva secretions [3,4].

Quantitative variations in salivary flow are defined as hyposalivation or hyposcialia and sialorrhea and may significantly impact the overall oral health and quality of life of affected individuals [5,6].

Hyposalivation, also known as dry mouth, is a multifactorial disorder characterized by a reduced or lack of normal salivary flow. It can arise from various etiologies, including medication side effects, systemic diseases such as Sjogren’s syndrome, radiation therapy, and dehydration [6,7]. Hyposalivation can cause a wide range of symptoms, including a dry, sticky feeling in the mouth, difficulty speaking or swallowing, altered taste, bad breath, and an increased risk of oral infections and tooth decay [7].

Recent evidence has reported that some patients reported individual’s experience of oral dryness, known as xerostomia. No underlying biological or functional disease is associated with this salivary complaint and the etiology may attributed to psychological factors including stress, anxiety, trauma or somatization although the exact cause remains not completely understood [7,8]. Similarly to hyposalivation, patients with xerostomia reported difficulty eating, speaking, and swallowing but also extraoral symptoms such as headaches, back pain, and gastrointestinal complaints have been reported [8,9].

On the other hand, excessive salivary production or pooling of saliva in the mouth is known as sialorrhea, or hypersalivation or drooling. It can occur due to various underlying causes, including neurological conditions (such as Parkinson’s disease, cerebral palsy, or stroke), medication side effects, oral motor dysfunction, or anatomical abnormalities [10,11]. This condition poses challenges for both patients and healthcare providers, as it can lead socially embarrassing, skin irritation, wetting of clothes and increased risk of aspiration and subsequently respiratory complications [11,12].

While a term exists for the subjective sensation of oral dryness, there is currently no equivalent term for sialorrhea, which has historically been considered an objective condition, neglecting the subjective experience of individuals [13]. This lack of appropriate terminology for the subjective symptom of sialorrhea can lead to confusion among healthcare providers, impairing their understanding and management of this condition.

In previous studies, it has been consistently reported that patients affected by Burning Mouth Syndrome (BMS) often experience salivary complaints do not show hyposalivation or hypersalivation regardless of the mensuration methods but may suffer for xerostomia, as well as a subjective sensation of sialorrhea in association with the sensation of burning [7,9,14,15]. These oral symptoms may further complicate the clinical picture making difficult the diagnosis and may increase the discomfort experienced of affected individuals [16].

BMS is a complex chronic, idiopathic pain disorder characterized by a persistent burning/dysesthetic sensation in the oral cavity, lasting for more than three months in the absence of any local or systemic pathological changes often accompanied by taste alterations and oral discomfort [17]. The presence of xerostomia and subjective sensation of sialorrhea in BMS patients, sometimes simultaneously associated seems to be a contradictory association highlighting the intricate nature of BMS and emphasizes the need for a comprehensive understanding of the underlying mechanisms [7,16]. Exploring the relationship between xerostomia, sialorrhea, and the burning sensation in BMS can provide valuable insights into the pathophysiology of this syndrome and contribute to more effective diagnostic and therapeutic approaches.

The scientific aims:-to investigate the interplay between xerostomia, the subjective sensation of sialorrhea, and the sensation of burning in a large sample of 500 BMS patients.-to identify potential predictors of xerostomia and subjective sensation of sialorrhea in BMS patients, shedding light on the underlying factors contributing to these complex symptoms.

To best to knowledge, this is the first study that has analysed this topic in complete way complaint in order to provide valuable insight into the etiology and possible targeted treatment.

## 2. Materials and Methods

### 2.1. Patients Enrolment

This is a cross-sectional study, considering all data were evaluated at time zero; only the data related to topical therapy with clonazepam were prospectively recorded but do not fall within the analytical assessment of the study. All participants were recruited from the Oral Medicine Department at Federico II University of Naples between March 2019 and February 2022. The study received approval from the University of Naples Federico II Ethical Committee (Approval Number: 251/19, granted on 20 February 2019). The research methods adhered to the guidelines outlined in the Strengthening the Reporting of Observational Studies in Epidemiology (STROBE) for observational studies. Informed consent was obtained from each participant after providing a clear understanding of the study’s objectives and methods. During the initial visit, each patient underwent clinical questioning by the physician regarding their symptoms. In particular, they were asked to respond positively or negatively regarding the presence or absence of the following symptoms: Burning, Xerostomia, Sialorrhea, Dysgeusia, Globus, Itching, Intraoral foreign body sensation, Subjective Halitosis, Tingling, Occlusal Dysesthesia, Change in tongue morphology, Oral Dyskinesia, Dysosmia. Based on the patient’s response regarding the presence or absence of symptoms. The 500 patients were categorized into four groups based on their salivary symptoms: Burning (B), Burning and Xerostomia (BX), Burning and Sialorrhea (BS), and Burning and Xerostomia and Sialorrhea (BXS).

### 2.2. Inclusion and Exclusion Criteria

The study included patients aged 18 or older who met the criteria for BMS according to the International Classification of Orofacial Pain (ICOP 2020) 1st edition [17]. These criteria encompassed patients experiencing intraoral burning or dysesthetic sensations recurring daily for at least two hours per day over a period of more than three months, without any evident causative lesions during clinical examination and investigation. The pain experienced was characterized by a burning quality and localized superficially in the oral mucosa. The patients included in the study had normal blood test results, including blood count, blood glucose levels, glycated haemoglobin, serum iron, ferritin, and transferrin [17,18], unstimulated salivary flow higher than 0.1 mL/min and a stimulated flow higher than 0.5 mL/min [14] and the ultrasound imaging of the major salivary glands using high frequency (7–15 MHz) probes showed structural integrity and the absence of any visible abnormalities [19].

The patients were excluded if they suffered from diseases recognized as potential causative factors of BMS [20], were unable to comprehend or complete the questionnaires, had a history of psychiatric, neurological, or organic brain disorders, were undergoing treatment with psychotropic drugs or systemic medications potentially associated with oral symptoms, had a history of alcohol or substance abuse, had Obstructive Sleep Apnea Syndrome (OSAS), or had infectious, inflammatory, autoimmune diseases, or malignant tumors. In this study, a thorough evaluation was conducted to exclude any organic causes of hyposalivation in all BMS patients that reported salivary complaints. Multiple assess-ments were performed, including salivary flow evaluation, blood tests encompassing an immunological panel (Antinuclear Antibodies (ANA), anti-SS-A (anti-Ro) and anti-SS-B (anti-La), ultrasound imaging of the major salivary glands, brain MRI, and a meticulous clinical examination. These tests aimed to detect specific antibodies or markers associated with such conditions, providing valuable information for the exclusion of organic causes [21]. Moreover, we excluded patients suffering from muscle weakness or anatomical abnormalities leading to dysfunction in the coordination of the swallowing mechanism can lead to difficulties in swallowing, excess pooling of saliva in the oral cavity, and sialorrhea.

The challenge–dechallenge–rechallenge test was used in all subjects with a suspicion of an adverse drug reaction [22].

### 2.3. Clinical Assessment

Patients underwent a comprehensive history assessment, including sociodemographic profile, risk factors (alcohol and smoke use), body mass index (BMI), systemic diseases and medications; general examination, and a detailed examination of dental, gingival, and oral mucosal conditions.

In parallel, blood tests, including an immunological panel, were carried out to identify any systemic conditions or autoimmune disorders that could contribute to salivary gland dysfunction.

The clinical evaluation and symptom assessment took place in accordance with previous studies that provided a comprehensive view of the symptomatic pattern in these patients [16,23,24], and it included: disease onset (expressed in months), symptomatology variation during the day, improvement during eating, adherence to topical therapy with clonazepam, presence of any of the following symptoms burning, dysgeusia, globus, Itching, intraoral foreign body sensation, subjective halitosis, tingling, occlusal dysesthesia, change in tongue morphology, oral dyskinesia, dysosmia and their localization.

Pain severity, anxiety and depression levels, and sleep quality were evaluated using standardized battery tests: the intensity of pain was measured using the Visual Analogue Scale (VAS) [25], while the quality of pain was assessed using the short form of the McGill Pain Questionnaire (SF-MPQ) [26]. The VAS score is calculated by gauging the distance along the line connecting ‘no pain’ to the patient’s marked point, yielding scores between 0 and 10 (0 signifying no oral symptoms and 10 representing the most severe imaginable discomfort) [27]. The SF-MPQ score is derived from the sum of item scores (ranging from 0 to 45). There are no prede-fined critical thresholds for interpreting the scores, and akin to the MPQ, a higher score indicates more intense pain [27].

Depressive symptoms were evaluated using the Hamilton Depression Rating Scale (HAM-D) [28], and both somatic and psychic anxiety were measured using the Hamilton Anxiety Rating Scale (HAM-A) [29]. For the HAM-D assessment, a score exceeding 7 signifies a level of impairment. Scores falling within the range of 7 to 17 denote mild depression, while scores between 18 and 24 indicate moderate depression, and scores exceeding 24 indicate severe depression [30]. In the case of HAM-A, individual items are scored on a scale of 0 to 4, and a cumulative score below 17 suggests mild severity, a score of 18 to 24 suggests mild to moderate se-verity, and a score of 25 to 30 suggests moderate to severe severity [29]. The subjective sleep quality was assessed using the Pittsburgh Sleep Quality Index (PSQI) [31], and daytime sleepiness was measured using the Epworth Sleepiness Scale (ESS) [32]. PSQI global scores greater than 5 effectively differentiate individuals with poor sleep from those with good sleep, exhibiting high sensitivity (90–99%) and specificity (84–87%) [33]; the ESS score is determined by adding the values of eight items, spanning from 0 to 24, with a threshold of over 10 indicating notable daytime sleepiness [34].

### 2.4. Salivary Flow Evaluation

Salivary flow rates have been evaluated by experts at rest and upon stimulation following the standardised procedures according to previous validated studies [35,36]. The collection was performed in two tubes: the first tube for unstimulated saliva flow, measured at least 1 h after eating, with the patient at rest, in a quite room, for five minutes; and the second tube for stimulated saliva flow, collected after the patient chewed on a sugar-free, lemon-flavored candy to promote salivation, also for five minutes. The standard range for unstimulated saliva flow was set at ≤0.1 mL/min, while the stimulated saliva flow range was set ≤0.5 mL/min. Patients with an unstimulated salivary flow of less than 0.1 mL/min and a stimulated flow of less than 0.5 mL/min were diagnosed with hyposalivation [14,36].

### 2.5. Salivary Glands Imaging

Ultrasound imaging of the major salivary glands using high frequency (7–15 MHz) probes was performed to visualize the structural integrity and detect any visible abnormalities. This non-invasive technique allowed for a detailed examination of the salivary gland architecture, including the presence of any structural lesions or changes. Furthermore, ultrasounds helped us to exclude any acute inflammation, sialolithiasis or abscesses, affecting the salivary glands [19].

### 2.6. Neurological Assessment

A comprehensive neurological and neuroradiologic evaluation, throughout brain MRI images, has been performed to rule out any symptomatic correlation with Parkinson’s disease, cerebral palsy, Amyotrophic Lateral Sclerosis (ALS), stroke or other relevant conditions [37]. Moreover, we also analyzed neuroradiological images which can help in the diagnosis and evaluation of conditions affecting the salivary glands: tumors identification within the salivary glands, detect blockages or narrowing in the salivary ducts, and evaluate the overall structure and morphology of the salivary glands. Imaging findings can aid in determining the underlying cause of symptoms such as excessive salivation (sialorrhea) or sensation dry mouth (xerostomia) [10,38].

### 2.7. Data Analysis

In order to achieve a power test value (1-Beta) of at least 99% and a significance level of no more than 1%, the sample size required for the study was determined to be 500 pa-tients diagnosed with BMS. This sample size was determined based on an effect size of 0.65, which was obtained from a previous research study on ARWMCs. The calculations were performed using GPower software (v 3.1.9).

The R software (v. 4.1.2) was used to perform the statistical analysis. To analyse the socio-demographic and clinical characteristics of patients classified into the four groups, descriptive statistics, including means, standard deviations (SD), medians, and inter-quartile ranges (IQR) were measured. Pearson Chi Square or Fisher’s exact test was used to assess the significant clinical differences among the percentages of the four groups, depending on cell frequencies [39].

The non-parametric ANOVA procedure by Kruskal-Wallis was employed to test for any differences among the recorded medians values of the HAM-A, HAM-D, SF-MPQ, VAS, PSQI, ESS, and CGI in the four groups, considering the data were not normally distributed.

Multiple tests with the Bonferroni correction in all the analyses were performed to counteract the multiple comparisons issue.

The odds ratio (OR) of sociodemographic characteristics and risk factors (age, gender, education, marital status, employment status, smoker, alcohol and BMI), oral symptoms (dysgeusia, globus, tingling), evaluation of Psychological Profile (HAM-A; HAM-D), evaluation of Pain (VAS; SF-MPQ); evaluation of Sleep Quality (PSQI; ESS) among patients of each group have been calculated using unconditional logistic regression.

To obtain unadjusted coefficient estimations a sequential regression model analysis was performed including the predictors one by one. As a final step, a full model analysis was carried out by considering all the predictors simultaneously to estimate the adjusted coefficients.

## 3. Results

### 3.1. Demographic and Clinical Data

This study included 500 patients with Burning Mouth Syndrome (BMS); the initial number of the sample has decreased from 553 to 500 as 53 patients were excluded, of which 49 patients did not meet the inclusion criteria for the study (Appendix A), and 4 patients did not provide consent to participate in the study (Figure 1). The patients were subsequently categorized into four groups based on their reported salivary symptoms: Burning (B: 140 patients), Burning and Xerostomia (BX: 253 patients), Burning and Sialorrhea (BS: 49 patients), and Burning and Xerostomia and Sialorrhea (BXS: 58 patients). The patients included in the current study did not show any statistically significant differences among the groups considering the gender, age and education as shown in Table 1. There were females prevalence in all the groups and the only statistically significant difference found concern the family situation, with higher percentage of widowed in BX and BXS groups, 15.4% and 12.1% respectively. The mean BMI and the prevalence of smokers and alcohol consumers did not exhibit any significant statistical differences across the groups.

### 3.2. Systemic Evaluation and Symptoms Characteristics

In Table 2, the systemic comorbidities and medications were compared among the four groups, revealing a statistically significant variation in blood thinners intake (*p*: 0.002 **). BX patients had a higher percentage (8.7%) of blood thinners usage compared to the other groups (B: 1.4%, BS: 0%, BXS: 1.7%). Regarding disease onset and symptom patterns, no significant differences were observed among the groups (Table 3). However, the BX group exhibited a longer diagnostic delay, with an average disease onset of 32.6 ± 49.6 months. Most patients reported a continuous symptoms pattern, and a higher proportion of patients in the BXS group (41.4%) experienced symptom improvement during eating (*p*: 0.006 **).

Furthermore, significant differences were identified in the adherence to topical clonazepam treatment, with the highest adherence found in the BX group (76.7%), followed by BXS (63.8%), B (61.4%), and BS (61.2%) groups (*p*: 0.004 **). The response to topical treatment varied among the groups, with a higher percentage of improvement observed in the BS group (20.4%), followed by the BXS group (15.5%), BX group (17%), and B group (10.7%).

Considering the other oral symptoms referred, as showed in Table 4, dysgeusia and globus are the most reported symptoms by BX group, also showing statistically significant differences (118; 46.6%; *p*: 0.001 **; 109; 43.1%; *p*: 0.001 ** respectively). Moreover, the symptoms that showed more relevance, in all the groups, among those reported, were change in tongue morphology and intraoral body sensation. In particular, the intraoral foreign body sensation percentage was higher in the BXS group (27.6%) and BS (26.5%), while the change in tongue morphology was higher in BS (32.7%) and BXS (24.1%).

Statistically significant differences were found among 4 groups in the location of pain/burning symptoms (Table 4). Specifically, BX group reported more generalized burning (135; 53.4%; *p*: 0.001 **). In addition, the remaining 118 patients (46.6%) of BX group complained of widespread symptoms in several locations compared with other groups (*p*: 0.001 **).

Analysing the psychological profile pattern (Table 5) no differences were found in pain tests (VAS; SF-MPQ), anxiety and depression scales (HAM-A and HAM-D) and sleep quality (PSQI and ESS) despite all groups suffered from high levels of pain and presented mild anxiety and depression and sleep disturbance.

### 3.3. Regression Analysis

Table 6 presents the results of simultaneous multiple linear regression analyses conducted to predict the occurrence of symptoms in four different groups: B, BX, BS, and BXS. The first model aimed to investigate the contributions of demographic variables and risk factors to the symptoms. In the B group, smoking habit (OR = 1.66, *p*-value: 0.043 *) and BMI (OR = 0.94, *p*-value: 0.024 *) were found to be associated with the burning symptom. Interestingly, in the BS group, there was a moderate negative correlation with males (OR = 0.51, *p*-value: 0.047). Conversely, in the BXS group, only alcohol showed a positive association (OR = 3.52, *p*-value: 0.042). To expand the analysis, model 2 included additional symptoms (dysgeusia, globus, and tingling). This resulted in a significant increase in the R2 values for the B, BX, and BS groups, indicating improved prediction of symptoms (B group: DR2 = 4.6%; *p*-value < 0.001 **; BX group: DR2 = 1.6%; *p*-value < 0.013 *; BS group: DR2 = 4.6; *p*-value < 0.002 **). However, the addition of these symptoms did not significantly increase the R2 for the BXS group. Model 3, which specifically tested the impact of blood thinners, was conducted solely in the BX group. It demonstrated a significant increase in the R2 (BX: DR2 = 2.3%, *p*-value < 0.001 **). In the further models, the addition of pain variables (VAS and S-MPQ), of anxiety and depression (HAM-A and HAM-D), of quality of sleep and daytime sleepiness (PSQI and ESS) did not increase the R2 values in neither of the groups. The final full model (model 6 in B, BS, and BXS groups; model 7 in BX group) incorporated all variables simultaneously. It explained 5.8%, 4.8%, and 5.4% of the variance in the B, BX, and BS groups, respectively, with significant p-values (<0.001 **, <0.001 **, 0.041 *). Notably, this model did not yield significant results for the BXS group. In details, in the B group, BMI was identified as a negative predictor, while dysgeusia and globus were positive predictors of the burning symptom (OR = 0.93, *p*-value: 0.021; OR = 1.84, *p*-value: 0.006 **; OR = 2.12, *p*-value: 0.002 **, respectively). In the BX group, both globus and blood thinners were negative predictors (OR = 0.54, *p*-value: 0.002 *; OR = 0.14, *p*-value: 0.002 **, respectively). Lastly, in the BS group, male gender and tingling sensation were negative predictors (OR = 0.51, *p*-value: 0.049 *; OR = 0.24, *p*-value: <0.001 *). These findings provide valuable insights into the associations between various factors and symptom types within each of the four groups under study.

## 4. Discussion

Burning Mouth Syndrome (BMS) is a complex, puzzling and multifaceted disorder characterized by a range of specific symptoms, in addition to the predominant burning sensation [16,40]. While isolated burning is relatively rare, it is commonly accompanied by other symptoms including salivary complaints that can complicate the evaluation and diagnosis process for clinicians [7,9,14].

This study provides novel insights into the prevalence and association of burning, xerostomia, and subjective sialorrhea in a large sample of 500 patients affected by BMS. Xerostomia emerged as a common symptom because it was reported by 50.6% of the subjects (253) in conjunction with the burning sensation. In contrast, isolated oral burning was found to be present in only 28% (140) of the participants.

The prevalence of xerostomia in BMS varies across different studies but it is consistently reported as one of the most prevalent symptoms following the hallmark burning sensation with a prevalence rate range from 20% to 70% [7,9,16]. It is noteworthy, that some studies have identified a reduced unstimulated salivary flow rate but a normal stimulated flow [41,42]; others have reported changes in salivary biomarkers of inflammation and oxidative stress, suggesting a potential physiological basis of objective hyposalivation in BMS [43,44].

However, the majority of studies has consistently highlighted the subjective nature of xerostomia in these patients and it is primarily considered a perceived symptom linked to somatosensory dysfunction [7,9,45,46]. Aberrant sensory processing in the oral cavity can lead to altered perceptions of oral sensations, including the subjective feeling of dryness [45,47]. Furthermore, as suggested from the results of this study, the improvement of salivary complaints after clonazepam treatment provides additional support for the subjective nature of xerostomia in BMS.

The patient’s medical history was collected, and a comprehensive clinical examination was conducted in order to evaluate the oral health status. This examination helped to rule out any localized oral conditions, infections or dental issues suggestive of an impairment of salivary gland function.

Furthermore, this multidimensional approach to evaluate the presence of burning sensation and xerostomia, in the absence of any signs of oral infection, along with the diverse and intricate ways in which patients describe their pain, provide strong support for the diagnosis of BMS. These factors, combined with the onset of symptomatology and the presence of multiple additional symptoms, serve as valuable guiding points in reaching a proper diagnosis also avoiding diagnostic delay.

Additionally, literature evidence demonstrates that dysphagia is primarily associated with autoimmune diseases [48], and in these cases, it is deemed necessary for the patient to consume water or liquids during swallowing in this study. In this study it was observed that despite the majority of patients have not experienced significant symptoms improvement during eating, none of them spontaneously reported a constant need to consume water in order to facilitate swallowing food. The lack of reported dependency on continuous water intake for swallowing food further supports the idea that xerostomia is primarily associated with altered sensory processing rather than a physiological inability to swallow which is generally associated to real hyposalivation.

Examining the additional symptoms reported by patients, a higher prevalence of globus sensation (43.1%) was observed in the BX group, further complicating the clinical assessment because this symptom can often be erroneously attributed to hyposalivation by clinicians. However, it is important to note that the higher prevalence of globus sensation is not exclusive to BX group but is also observed in other groups. Therefore, similarly to burning sensation, xerostomia, globus and others addition symptoms may be related to central and peripheral neuropathy rather than being caused by organic conditions.

Considering the fact that a high percentage of patients reported the perception of morphological alterations of the tongue in the absence of clinical signs, respectively: B: 20 (14.3%), BX: 58 (22.9%), BS: 16 (32.7%), BXS: 14 (24.1%), this data can assist the clinician in navigating the diagnosis of exclusion and optimizing the specific therapeutic assessment for the patient. Indeed, the actual presence of qualitative or quantitative salivary alteration should be associated with effective morphological variation of the oral mucosa, particularly of the tongue [49].

Despite the precise mechanisms underlying the connection between burning and xerostomia are not fully elucidated the higher prevalence of xerostomia in patients with small fiber neuropathy may support the neuropathic pathogenesis of BMS [47,50]. Indeed, the perception or amplification of xerostomia may be attributed to a dysfunction or damage in the peripheral or central nervous system, which can result in altered sensory processing and perception of various sensations [7,45]. Specifically, the dysfunction or damage of sensory nerve fibers responsible for regulating the salivary gland functions and perceiving oral moisture may play a role in the development of xerostomia symptoms in these individuals [47,51]. Furthermore, central sensitization involving an amplification of pain or sensory signals within the central nervous system, including xerostomia, can lead to an increased perception of discomfort [52,53].

In this study, an interesting finding was observed in the BX group, where a higher percentage of patients reported using blood thinners (8.7%); in addition, multiple correlation analysis revealed that these drugs were identified as a negative predictor of xerostomia in this subset of patients. However, it is widely recognized that the use of blood thinners does not directly influence saliva production even if few studies reported an association between the use of blood thinners and the occurrence of xerostomia [54,55,56]. It is crucial to consider that blood thinner are life-saving medications commonly prescribed to individuals at risk of blood clots, for prevention of stroke, heart attack, or pulmonary embolism therefore in the majority of the cases is impossible to stop, replace or introduce the drug considering that the occurrence of xerostomia is a relatively uncommon side effect compared to the overall benefits of these medications [57,58].

In this study, an association between the burning sensation and the presence of subjective sialorrhea has been reported. Specifically, 9.8% (49) of the participants experienced both burning and sialorrhea simultaneously. Moreover, we identified a similar pattern with respect to the co-occurrence of burning, subjective sialorrhea, and xerostomia, as 11.6% (58) of the subjects reported experiencing all the three symptoms together. The presence of this symptom occurring alone or in conjunction with xerostomia, poses additional complexity for clinicians and significantly hampers the diagnostic process. It becomes essential to differentiate between objective and subjective sialorrhea, even though there is currently no specific term to describe subjective sialorrhea [11,13].

Clinicians rely on the patient’s subjective experience, history, and symptom description to differentiate between objective and subjective sialorrhea [12,13]. In order to reach an accurate diagnosis, it is crucial to exclude potential underlying neurological disorders that can disrupt the coordination of the muscles involved in swallowing [36,38]. Conditions such as Parkinson’s disease, cerebral palsy, or amyotrophic lateral sclerosis are known to affect the normal functioning of these muscles, leading to the accumulation of saliva in the mouth [38,59,60]. Other factors that can contribute to sialorrhea include facial nerve paralysis or oral motor dysfunction, as well as the use of medications such as antipsychotics, anticholinergics, or antiemetics [5].

To ensure a comprehensive evaluation, all patients in this study underwent MRI scans of the brain to exclude the presence of neurodegenerative diseases. Additionally, in selected cases, patients underwent neurological assessments to further investigate any potential underlying neurological causes. The thorough exclusion of organic causes plays a crucial role in guiding clinicians towards the diagnosis of subjective sialorrhea [61]. When organic causes are effectively ruled out, it becomes plausible to consider that the patient’s subjective experience of excessive saliva may be attributed to altered sensory perception, which can be further amplified by anxiety or psychological stress [62].

In these instances, a mismatch arises between the patient’s perceived saliva production and their ability to effectively manage and swallow saliva. The altered sensory processing in the central nervous system may lead to an exaggerated perception of saliva production, causing the patient to perceive excessive sialorrhea despite objective measurements indicating normal saliva production [63,64].

Moreover, in this study, the presence of salivary complaint has been associated with a higher prevalence of additional symptoms, including dysgeusia, globus, tingling, and dysperceptive symptoms that involved the entire oral cavity. These symptoms collectively impact on the complexity of the BMS management; particularly the higher prevalence of dysgeusia, can further contribute to the patient’s dissatisfaction and discomfort, as it affects their ability to enjoy food and beverages that in the time may worsening psychological profile and quality of life of the patients [65].

The results of multiple regression analysis conducted in this study provide some insight into the potential predictors even if these results could not explain the complexity of the symptomatology in BMS patients. In the B group, several factors, including smoking, BMI, dysgeusia, globus sensation, and tingling sensations, were identified as predictors of the burning sensation. Similarly, in the BX group, globus sensation, the use of blood thinners, and daytime sleepiness were associated with both burning and xerostomia. In the BS group, male gender and tingling sensations were predictive of burning and sialorrhea. Lastly, in the BXS group, alcohol consumption, dysgeusia, and the quality of pain were predictors of burning, xerostomia, and sensation of sialorrhea.

However, when all these variables were considered together in the final full models, they could only explain a small percentage of the variance in the symptoms. Specifically, the models accounted for 5.8% of the variance in burning, 4.8% in burning and xerostomia, 5.4% in burning and sialorrhea, and 3% in burning, xerostomia, and sialorrhea combined. These findings suggest that there are other unidentified factors that contribute to the manifestation of specific symptoms in BMS.

Interestingly anxiety, depression and sleep disturbance were not identified as predictors of the symptoms in this study. However, it is well-established that psychological factors, including anxiety and stress, can significantly exacerbate the perception of subjective saliva-related sensations [43,62]. It is not possible to exclude that mood disorders can intensify the patient’s focus on the subjective experience of excess or reduced saliva production [44]. The mind-body connection plays a vital role in these cases, as the patient’s psychological state can greatly influence their perception and interpretation of sensory signals associated with saliva production [62]. Indeed, in this study all the patients exhibited high levels of anxiety, depression, and poor sleep quality, which can further contribute to heightened attention towards bodily symptoms.

This study has also several limitations: firstly, it was not possible to understand if the salivary complaints was antecedent to the burning symptomatology or it appears after; moreover, we did not collect information regarding the timing and sequence of the salivary symptoms occurrence. Secondarily, the limited explanatory power of the regression models underscores the complex nature of BMS, characterized by a multitude of factors contributing to salivary complaints in the absence of identifiable organic pathology. This highlights the need for continued research to unravel the underlying mechanisms driving BMS symptomatology.

## 5. Conclusions

The results of the present study have highlighted the significant variability in symptomatology reported by patients affected by BMS, particularly concerning salivary complaints. Specifically oral burning frequently is associated with xerostomia and sialorrhea, which may also overlap.

The complexity in interpreting salivary symptoms in these patients, even if purely subjective, requires careful assessment for evaluating salivary flow. Moving forward, a comprehensive research effort should be considered in the development of specific diagnostic criteria and assessment tools capable of excluding any organic nature of the symptoms and subsequently to differentiate between objective and subjective hyposalivation symptoms.

The development of specific diagnostic algorithm could aid clinicians in making timely and accurate diagnoses ultimately leading to improved patient care and treatment outcomes. By addressing individual and unique patient’s symptomatology, clinicians can tailor personalized management strategies to an enhance the quality of life and overall well-being of BMS patients.

## Figures and Tables

**Figure 1 jcm-12-05561-f001:**
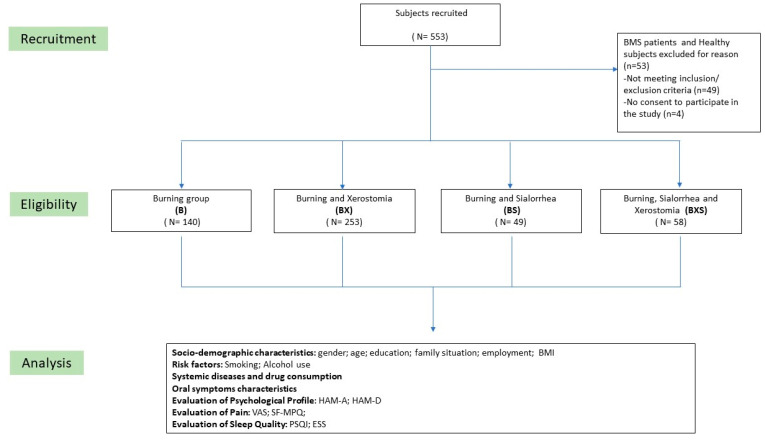
Flow Chart of the study. Abbreviations: BMS: Burning Mouth Syndrome; BMI: Body Mass Index; HAM-A: Hamilton rating scale for anxiety; HAM-D: Hamilton rating scale for depression; VAS: Visual Analogue Scale; SF-MPQ: Short Form Mc-Gill Pain Questionnaire; PSQI: Pittsburgh Sleep Quality Index; ESS: Epworth Sleepiness Scale.

**Table 1 jcm-12-05561-t001:** Sociodemographic characteristics and Risk factors in the four groups: Burning (B: 140), Burning + Xerostomia (BX: 253), Burning + Sialorrhea (BS: 49), Burning + Xerostomia + Sialorrhea (BXS: 58).

Demographic Variables	B (140)	BX (253)	BS (49)	BXS (58)	*p*-Value
	Frequency %	Frequency %	Frequency %	Frequency %	
Gender - Male - Female	39 (27.9) 101 (72.1)	3 (24.9) 190 (75.1)	18 (36.7) 31 (63.3)	9 (15.5) 49 (84.5)	0.082
Employment - Employed - Unemployed - Retired	30 (21.4) 71 (50.7) 39 (27.9)	58 (22.9) 110 (43.5) 85 (33.6)	14 (28.6) 16 (32.7) 19 (38.8)	17 (29.3) 20 (34.5) 21 (36.2)	0.447
Family situation - Single - Married - Divorced - Widowed	10 (7.1) 114 (81.4) 7 (5) 9 (6.4)	17 (6.7) 188 (74.3) 9 (3.6) 39 (15.4)	8 (16.3) 34 (69.4) 4 (8.2) 3 (6.1)	2 (3.4) 45 (77.6) 4 (6.9) 7 (12.1)	0.038 *
	Mean ± SD	Mean ± SD	Mean ± SD	Mean ± SD	*p*-value
Age (in years)	64 ± 12.7	65.4 ± 12.5	61.3 ± 15.4	64 ± 12.9	0.193
Education (in years)	9.33 ± 4.70	9.07 ± 4.51	9.43 ± 4.37	9.28 ± 4.59	0.929
Body Mass Index	27.4 ± 3.38	26.7 ± 3.53	26.3 ± 2.61	26.1 ± 4.16	0.059
Risk factors	Frequency (%)	Frequency (%)	Frequency (%)	Frequency (%)	*p*-value
Smoker - cigarettes < 5 - cigar. 5–10 - cigar. 10–15 - cigar. >15 - Not smoker	5 (3.6) 6 (4.3) 10 (7.1) 7 (5) 112 (80)	12 (4.7) 11 (4.3) 16 (6.3) 25 (9.9) 189 (74.7)	3 (6.1) 0 (0) 4 (8.2) 11 (22.4) 31 (63.3)	1 (1.7) 3 (5.2) 10 (17.2) 6 (10.3) 38 (65.5)	0.148
Alcohol use - 1–2 units - 2–3 units - >3 units - No Alcohol	15 (10.7) 4 (2.9) 2 (1.4) 119 (85)	27 (10.7) 12 (4.7) 4 (1.6) 210 (83)	6 (12.2) 1 (2) 0 (0) 42 (85.7)	3 (5.2) 0 (0) 0 (0) 55 (94.8)	0.755

The significance difference among the percentages was measured by the Pearson Chi Square test. * Significant 0.01 < *p* ≤ 0.05.

**Table 2 jcm-12-05561-t002:** Systemic Comorbidities and medications in the four groups: Burning (B: 140), Burning + Xerostomia (BX: 253), Burning + Sialorrhea (BS: 49), Burning + Xerostomia + Sialorrhea (BXS: 58).

	B	BX	BS	BXS	*p*-Value
Comorbidities	Frequency (%)	Frequency (%)	Frequency (%)	Frequency (%)	
Total	107 (76.4)	219 (86.6)	45 (91.8)	49 (84.5)	0.029
Hypertension	65 (46.4)	131 (51.8)	26 (53.1)	27 (46.6)	0.683
Hypercholesterolemia	56 (40)	95 (37.5)	20 (40.8)	22 (37.9)	0.949
Myocardial infarction	6 (4.3)	10 (4)	2 (4.1)	2 (3.4)	1.000
Other Cardiovascular diseases	8 (5.7)	23 (9.1)	5 (10.2)	5 (8.6)	0.604
Respiratory diseases	3 (2.1)	12 (4.7)	2 (4.1)	0 (0)	0.245
Gastrointestinal diseases	23 (16.4)	49 (19.4)	8 (16.3)	10 (17.2)	0.900
Endocrine diseases	0 (0)	4 (1.6)	3 (6.1)	0 (0)	0.025
Prostatic Hypertrophy	4 (2.9)	17 (6.7)	2 (4.1)	3 (5.2)	0.433
Hypothyroidism	14 (10)	42 (16.6)	7 (14.3)	5 (8.6)	0.207
Hyperthyroidism	3 (2.1)	0 (0)	1 (2)	1 (1.7)	0.047
HCV infection	3 (2.1)	4 (1.6)	2 (4.1)	0 (0)	0.400
HBV infection	1 (0.7)	2 (0.8)	0 (0)	0 (0)	1.000
Neoplastic disease	7 (5)	16 (6.3)	1 (2)	6 (10.3)	0.341
Neurological disease	5 (3.6)	2 (0.8)	1 (2)	0 (0)	0.146
Others	16 (11.4)	60 (23.7)	6 (12.2)	14 (24.1)	0.009
Medications	Frequency (%)	Frequency (%)	Frequency (%)	Frequency (%)	*p*-value
Total Drugs intake	85 (60.7)	193 (76.3)	32 (65.3)	38 (65.5)	0.009
ACE-inhibitors	19 (13.6)	33 (13)	4 (8.2)	6 (10.3)	0.793
Calcium Channel blockers	12 (8.6)	25 (9.9)	1 (2)	7 (12.1)	0.244
Angiotensin II receptor antagonists	23 (16.4)	40 (15.8)	12 (24.5)	9 (15.5)	0.508
Thiazide Diuretics	10 (7.1)	41 (16.2)	7 (14.3)	5 (8.6)	0.049
Beta blockers	21 (15)	45 (17.8)	5 (10.2)	9 (15.5)	0.620
Statins	36 (25.7)	66 (26.1)	11 (22.4)	12 (20.7)	0.845
Ezetimibe	0 (0)	9 (3.6)	1 (2)	0 (0)	0.057
Antiplatelets	40 (28.6)	76 (30)	14 (28.6)	15 (25.9)	0.944
Blood thinners	2 (1.4)	22 (8.7)	0 (0)	1 (1.7)	0.002 **
Bisphosphonates	4 (2.9)	5 (2)	1 (2)	1 (1.7)	0.960
Levothyroxin sodium	12 (8.6)	33 (13)	5 (10.2)	8 (13.8)	0.545
Steroids	2 (1.4)	7 (2.8)	0 (0)	3 (5.2)	0.291
Proton Pump inhibitors	29 (20.7)	65 (25.7)	10 (20.4)	17 (29.3)	0.491
Others	16 (11.4)	39 (15.4)	4 (8.2)	7 (12.1)	0.515

A significance difference between the percentages was measured by the Pearson Chi Square test. When one or more cells contain a frequency less than 5 then the Fisher Exact Test was used. ** Significant with Bonferroni correction 0.003.

**Table 3 jcm-12-05561-t003:** Disease onset, Symptoms pattern and topical therapy response in the four groups: Burning (B: 140), Burning + Xerostomia (BX: 253), Burning + Sialorrhea (BS: 49), Burning + Xerostomia + Sialorrhea (BXS: 58).

	B	BX	BS	BXS	*p*-Value
	Mean ± SD	Mean ± SD	Mean ± SD	Mean ± SD	
Disease onset (months)	26.4 ± 47.9	32.6 ± 49.6	34.5 ± 53.2	20.8 ± 21.8	0.236
*Symptoms pattern*	Frequency (%)	Frequency (%)	Frequency (%)	Frequency (%)	
Worst in the morning	3 (2.1)	12 (4.7)	1 (2)	1 (1.7)	0.583
Worst in the afternoon/evening	55 (39.3)	104 (41.1)	17 (34.7)	29 (50)	0.412
Same morning/afternoon/evening	79 (56.4)	134 (53)	30 (61.2)	28 (48.3)	0.527
Continuous	90 (64.3)	152 (60.1)	32 (65.3)	33 (56.9)	0.690
Intermittent	49 (35)	92 (36.4)	16 (32.7)	25 (43.1)	0.677
Improvement during eating	26 (18.6)	58 (22.9)	15 (30.6)	24 (41.4)	0.006 **
*Topical Therapy*	Frequency (%)	Frequency (%)	Frequency (%)	Frequency (%)	
Topical Clonazepam	86 (61.4)	194 (76.7)	30 (61.2)	37 (63.8)	0.004 **
Improvement with topical treatment	15 (10.7)	43 (17)	10 (20.4)	9 (15.5)	0.262

A significance difference between the percentages was measured by the Pearson Chi Square test. When one or more cells contain a frequency less than 5 then the Fisher Exact Test was used. ** Significant with Bonferroni correction 0.005 for the disease onset.

**Table 4 jcm-12-05561-t004:** Oral symptoms and localization in the four groups: Burning (B: 140), Burning + Xerostomia (BX: 253), Burning + Sialorrhea (BS: 49), Burning + Xerostomia + Sialorrhea (BXS: 58).

	B	BX	BS	BXS	*p*-Value
*Oral Symptoms*	Frequency (%)	Frequency (%)	Frequency (%)	Frequency (%)	
Burning	140 (100)	253 (100)	49 (100)	58 (100)	-
Dysgeusia	45 (32.1)	118 (46.6)	28 (57.1)	34 (58.6)	0.001 **
Globus	32 (22.9)	109 (43.1)	19 (38.8)	23 (39.7)	0.001 **
Itching	16 (11.4)	28 (11.1)	10 (20.4)	7 (12.1)	0.339
Intraoral foreign body sensation	27 (19.3)	50 (19.8)	13 (26.5)	16 (27.6)	0.392
Subjective Halitosis	5 (3.6)	17 (6.7)	4 (8.2)	4 (6.9)	0.449
Tingling	10 (7.1)	29 (11.5)	14 (28.6)	5 (8.6)	0.002 **
Occlusal Dysesthesia	7 (5)	19 (7.5)	11 (22.4)	4 (6.9)	0.006
Change in tongue morphology	20 (14.3)	58 (22.9)	16 (32.7)	14 (24.1)	0.033
Oral Dyskinesia	7 (5)	20 (7.9)	8 (16.3)	4 (6.9)	0.109
Dysosmia	2 (1.4)	12 (4.7)	6 (12.2)	5 (8.6)	0.009
*Location of Pain/Burning*	Frequency (%)	Frequency (%)	Frequency (%)	Frequency (%)	*p*-value
Generalized	35 (25%)	135 (53.4%)	22 (44.9%)	36 (62.1%)	<0.001 **
Gums	61 (43.6)	159 (62.8)	33 (67.3)	42 (72.4)	<0.001 **
Lips	56 (40)	180 (71.1)	30 (61.2)	45 (77.6)	<0.001 **
Cheeks	43 (30.7)	157 (62.3)	25 (51)	40 (69)	<0.001 **
Tongue	119 (85)	241 (95.3)	43 (87.8)	53 (91.4)	0.004 **
Floor of the mouth	42 (30)	140 (55.3)	25 (51)	35 (60.3)	<0.001 **
Palate	63 (45)	175 (69.2)	25 (51)	42 (72.4)	<0.001 **
Trigone	37 (26.4)	131 (51.8)	19 (38.8)	40 (69)	<0.001 **

A significance difference between the percentages was measured by the Pearson Chi Square test. When one or more cells contain a frequency less than 5 then the Fisher Exact Test was used. ** Significant with Bonferroni correction 0.003 for the Oral Symptoms. ** Significant with Bonferroni correction 0.006 for the location of pain/burning.

**Table 5 jcm-12-05561-t005:** Psychological profile and Sleep evaluation in the four groups: Burning (B: 140), Burning + Xerostomia (BX: 253), Burning + Sialorrhea (BS: 49), Burning + Xerostomia + Sialorrhea (BXS: 58).

	B	BX	BS	BXS	*p*-Value
	Median (IQR)	Median (IQR)	Median (IQR)	Median (IQR)	Median (IQR)
** *VAS* **	10 [8.75–10]	10 [9–10]	10 [9–10]	10 [9.25–10]	0.288
** *SF-MPQ* **	10.5 [7.75–12]	10 [7–12]	9 [7–13]	11 [9–12]	0.168
** *HAM-A* **	16.5 [15–20]	18 [15–20]	17 [15–21]	18 [16–21]	0.276
** *HAM-D* **	16 [14–19]	17 [14–20]	17 [14–20]	16.5 [14–21.75]	0.785
** *PSQI* **	8 [8–10]	8 [7–10]	8 [8–10]	8 [8–10]	0.968
** *ESS* **	7 [5–9]	7 [5–9]	8 [5–9]	7 [5–9]	0.337

A significance difference between the percentages was measured by the Kruskall-Wallis test. ** Significant with Bonferroni correction 0.007. Abbreviations: VAS, Visual Analogue Scale; SF-MPQ, Short-Form McGill Pain Questionnaire; HAM-A, Hamilton rating scale for anxiety; HAM-D, Hamilton rating scale for depression; PSQI, Pittsburgh sleep quality index; ESS, Epworth Sleepiness Scale.

**Table 6 jcm-12-05561-t006:** Multivariate logistic regression analysis in the four groups considering sociodemographic variables and risk factors (Model 1); dysgeusia, globus and tingling (Model 2); comorbidities, drugs intake, antiplatelets (Model 3); VAS and SF-MPQ (Model 4); HAM-A and HAM-D (Model 5); PSQI and ESS (Model 6); all the variables (Model 7).

BURNING (B)	Model 1			Model 2			Model 3		Model 4			Model 5			Model 6		
	*OR*		*p-Value*	*OR*	*p-Value*		*OR*	*p-Value*	*OR*	*p-Value*		*OR*	*p-Value*		*OR*	*p-Value*	
Age	1.01		0.292	1.01	0.219		1.01	0.272	1.01	0.289		1.01	0.290		1.01	0.188	
Gender: Male	0.89		0.625	0.99	0.976		0.87	0.552	0.89	0.608		0.88	0.584		0.95	0.838	
Years of education	0.99		0.719	0.99	0.715		1.00	0.888	0.99	0.734		0.99	0.813		1.00	0.924	
Marital status: Married	0.69		0.136	0.65	0.095		0.70	0.153	0.69	0.138		0.69	0.147		0.64	0.092	
Job: Occupied	1.19		0.529	1.25	0.434		1.20	0.514	1.19	0.530		1.24	0.449		1.30	0.368	
Smoker	1.66		0.043 *	1.53	0.098		1.65	0.045 *	1.64	0.049 *		1.60	0.059		1.48	0.130	
Alcohol	0.89		0.687	0.86	0.628		0.88	0.678	0.90	0.715		0.94	0.827		0.91	0.756	
BMI	0.94		0.024 *	0.94	0.027 *		0.93	0.018 *	0.93	0.020 *		0.94	0.032 *		0.93	0.021 *	
Dysgeusia				1.87	0.004 **										1.84	0.006 **	
Globus				2.17	0.001 **										2.12	0.002 **	
Tingling				1.67	0.178										2.02	0.077	
VAS							1.17	0.061							1.17	0.073	
SF-MPQ							1.01	0.579							1.00	0.907	
HAM-A									1.02	0.492					1.03	0.404	
HAM-D									0.99	0.705					0.97	0.362	
PSQI												1.04	0.378		1.03	0.554	
ESS												0.94	0.096		0.94	0.109	
*R*^2^ *(%)*	2.3		0.086	6.9	<0.001 **		3	0.059	2.4	0.157		2.9	0.076		8.1	<0.001 **	
*R*^2^ *change (%)*				4.6	<0.001 **		0.7	0.140	0.1	0.773		0.6	0.217		5.8	<0.001 **	
**BURN + XEROSTOMIA (BX)**	**Model 1**		**Model 2**		**Model 3**		**Model 4**		**Model 5**			**Model 6**			**Model 7**		
	*OR*	*p-value*	*OR*	*p-value*	*OR*	*p-value*	*OR*	*p-value*	*OR*	*p-value*		*OR*	*p-value*		*OR*	*p-value*	
Age	1.45	0.128	0.99	0.098	0.99	0.278	0.99	0.139	0.99	0.117		0.99	0.117		0.99	0.209	
Gender: Male	1.19	0.414	1.12	0.614	1.30	0.230	1.21	0.367	1.20	0.384		1.20	0.405		1.26	0.296	
Years of education	1.00	0.824	1.00	0.873	1.01	0.752	1.00	0.984	1.01	0.784		1.00	0.915		1.00	0.978	
Marital status: Married	1.21	0.362	1.27	0.272	1.23	0.346	1.23	0.338	1.22	0.357		1.21	0.378		1.31	0.227	
Job: Occupied	0.86	0.540	0.83	0.471	0.80	0.384	0.84	0.492	0.87	0.577		0.84	0.481		0.76	0.294	
Smoker	1.03	0.878	1.10	0.670	0.99	0.944	1.05	0.826	1.03	0.884		1.07	0.751		1.09	0.694	
Alcohol	0.71	0.204	0.72	0.212	0.75	0.301	0.70	0.187	0.71	0.197		0.68	0.153		0.71	0.226	
BMI	1.01	0.733	1.01	0.736	1.01	0.714	1.01	0.702	1.01	0.627		1.00	0.851		1.01	0.702	
Disgeusia			0.97	0.849											0.98	0.908	
Globus			0.53	0.002 **											0.54	0.002 **	
Tingling			1.18	0.576											1.11	0.739	
Blood thinners					0.13	0.001 **									0.14	0.002 **	
VAS							0.91	0.219							0.92	0.341	
SF-MPQ							1.01	0.416							1.02	0.223	
HAM-A									0.97	0.229					0.97	0.289	
HAM-D									1.04	0.204					1.04	0.160	
PSQI												0.98	0.605		0.97	0.533	
ESS												1.07	0.041 *		1.07	0.065	
*R*^2^ *(%)*	0.9	0.583	2.5	0.100	3.2	0.009 **	1.2	0.567	1.2	0.595		1.6	0.373		5.7	0.003 **	
*R*^2^ *change (%)*			1.6	0.013 *	2.3	<0.001 **	0.3	0.356	0.3	0.411		0.7	0.120		4.8	<0.001 **	
**BURN + SIALORRHEA** **(BS)**	**Model 1**			**Model 2**			**Model 3**		**Model 4**			**Model 5**			**Model 6**		
	*OR*		*p-value*	*OR*	*p-value*		*OR*	*p-value*	*OR*		*p-value*	*OR*		*p-value*	*OR*		*p-value*
Age	1.02		0.174	1.02	0.190		1.02	0.173	1.02		0.165	1.02		0.165	1.02		0.181
Gender: Male	0.51		0.047 *	0.49	0.036		0.53	0.056	0.51		0.042 *	0.51		0.047 *	0.51		0.049 *
Years of education	1.01		0.758	1.02	0.685		1.01	0.838	1.01		0.741	1.01		0.737	1.01		0.762
Marital status: Married	1.42		0.302	1.40	0.336		1.41	0.306	1.41		0.306	1.42		0.301	1.39		0.343
Job: Occupied	1.05		0.908	1.04	0.931		1.02	0.950	1.03		0.936	1.05		0.910	0.98		0.964
Smoker	0.62		0.142	0.64	0.197		0.62	0.144	0.61		0.131	0.61		0.135	0.62		0.161
Alcohol	1.23		0.654	1.18	0.717		1.21	0.678	1.23		0.647	1.23		0.646	1.15		0.761
BMI	1.05		0.310	1.06	0.255		1.05	0.288	1.04		0.367	1.05		0.299	1.06		0.261
Disgeusia				0.57	0.079										0.58		0.094
Globus				1.25	0.512										1.22		0.556
Tingling				0.26	<0.001 **										0.24		<0.001 **
VAS							0.88	0.386							0.82		0.213
SF-MPQ							1.00	0.965							1.01		0.772
HAM-A									1.06		0.281				1.04		0.429
HAM-D									0.96		0.421				0.99		0.827
PSQI												1.00		0.984	0.99		0.885
ESS												0.97		0.638	0.97		0.648
*R*^2^ *(%)*	3.5		0.196	8.1	0.007 **		3.7	0.290	3.8		0.267	3.5		0.331	8.9		0.038 *
*R*^2^ *change (%)*				4.6	0.002 **		0.2	0.664	0.3		0.554	0.0		0.886	5.4		0.041 *
**BURN + XERO + SIAL** **(BXS)**	**Model 1**			**Model 2**			**Model 3**		**Model 4**			**Model 5**			**Model 6**		
	*OR*		*p-value*	*OR*	*p-value*		*OR*	*p-value*	*OR*		*p-value*	*OR*		*p-value*	*OR*		*p-value*
Age	0.99		0.562	0.99	0.598		0.99	0.502	0.99		0.615	0.99		0.583	0.99		0.600
Gender: Male	1.68		0.183	1.69	0.181		1.64	0.205	1.69		0.178	1.70		0.175	1.67		0.196
Years of education	0.99		0.829	0.99	0.781		1.00	0.909	0.99		0.749	0.99		0.817	0.99		0.840
Marital status: Married	0.88		0.706	0.89	0.728		0.85	0.629	0.88		0.698	0.88		0.702	0.85		0.648
Job: Occupied	0.97		0.928	0.94	0.868		1.02	0.954	0.94		0.876	0.97		0.928	0.99		0.987
Smoker	0.61		0.11	0.59	0.092		0.58	0.087	0.64		0.155	0.60		0.108	0.58		0.090
Alcohol	3.52		0.042 *	3.48	0.044 *		3.70	0.035 *	3.42		0.047 *	3.52		0.043 *	3.65		0.039 *
BMI	1.07		0.096	1.07	0.120		1.08	0.079	1.07		0.101	1.08		0.090	1.08		0.103
Dysgeusia				0.53	0.031 *										0.56		0.052
Globus				1.01	0.977										1.02		0.941
Tingling				1.56	0.377										1.85		0.239
VAS							1.00	0.973							1.05		0.733
SF-MPQ							0.95	0.045 *							0.95		0.049 *
HAM-A									0.99		0.862				1.00		0.927
HAM-D									0.97		0.548				0.97		0.428
PSQI												0.98		0.751	1.01		0.861
ESS												0.97		0.537	0.96		0.468
*R*^2^ *(%)*	4.0		0.076	5.5	0.049 *		5.0	0.054	4.3		0.118	4.1		0.139	7.0		0.093
*R*^2^ *change (%)*				1.5	0.136		1.0	0.148	0.3		0.554	0.1		0.740	3.0		0.282

The *p*-values were obtained from the hypothesis test on regression coefficients. * Moderately significant 0.01 < *p*-value ≤ 0.05. ** Strongly significant *p*-value ≤ 0.01. Abbreviations: BMI, Body Mass Index; VAS, Visual Analogue Scale; SF-MPQ, Short-Form McGill Pain Questionnaire; HAM-A, Hamilton rating scale for anxiety; HAM-D, Hamilton rating scale for depression; PSQI, Pittsburgh sleep quality index; ESS, Epworth Sleepiness Scale.

## Data Availability

The data that support the findings of this study are available from the corresponding author, G.S., upon reasonable request.

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
