# Peer review of "Salivary Complaints in Burning Mouth Syndrome: A Cross Sectional Study on 500 Patients"

_jcm, 2023, doi:10.3390/jcm12175561_

Round 1

Reviewer 1 Report

The research is interesting, it focus on a clinically relevant issue for the management of patients with Burning Mouth Syndrome. The work raises novel points of view. Among them, a difficult question to resolve: how the difference between the sensation of a dry mouth or a “wet mouth" and the objective production of saliva (hyposialia / sialorrhea) within the context of BMS can be managed.

We consider that the work is ambitious, but it presents several problems that must be solved: The objectives of the study include "to identify  potential predictors of xerostomia and subjective sensation of sialorrhea". The subjective sensation could be identified against the actual production of saliva if the results were compared. We have not found in the paper a correlation between the sensation related to saliva reported by the patient and the current production of saliva. The relationship between salivary production and the sensation of salivary production is unknown, so it cannot be said that the sensation is subjective, it means that you can not achieved your main objective.

The exclusion criteria exclude organic neurological pathologies related to BMS. But psychological disturbances related to BMS are accepted, as well as hypertension or thyroid diseases and treatments for these diseases or even antineoplastic treatments that are known to be related to BMS symptoms and saliva production. At no time in the selection of the sample was specified that the patients were being treated for BMS. The different treatments are not described whether there was a clearance period or how long they had been on treatment. It must be explained.

In the conclusions, the improvement of xerostomia is related to topical treatment, but it is not related to the production of saliva again, so it cannot be concluded if it is about xerostomia or hyposialia.

Besides these problems it has not been specified why non-parametircs tests are used for the evaluation of statistical significance.

Based on these problems of sample selection and evaluation of results, the conclusions reached cannot be considered entirely valid. We believe that these aspects should be changed or at least clarified in order to accept the job.

Author Response

The research is interesting, it focus on a clinically relevant issue for the management of patients with Burning Mouth Syndrome. The work raises novel points of view. Among them, a difficult question to resolve: how the difference between the sensation of a dry mouth or a “wet mouth" and the objective production of saliva (hyposialia / sialorrhea) within the context of BMS can be managed.

Thank you for your comment. The difference between the sensation of a dry mouth or a "wet mouth" and the objective production of saliva (hyposialia/sialorrhea) within the context of Burning Mouth Syndrome (BMS) can be challenging but essential for providing effective treatment and relief to patients. In the discussion paragraph we have underlined the importance of the clinical assessment (medical and dental history, oral health status), including a physical examination to assess salivary gland function. In addition, underlying causes including oral infections, hormonal imbalances, and certain medications have been evaluated in this study and they are part of our exclusion criteria.

In this study we have evaluated several salivary function tests, in order to identify the potential causes of hyposialia or sialorrhea. However, no studies have reported which are the suitable tests able to evaluate the salivary flow in this type of patients so far. This study may be a starting point in the development of new clinical guidelines for the assessment of these patients. In the next future, it could be useful to develop a specific symptom questionnaire in order to let the patients communicate their subjective experience accurately.

We consider that the work is ambitious, but it presents several problems that must be solved: The objectives of the study include "to identify potential predictors of xerostomia and subjective sensation of sialorrhea". The subjective sensation could be identified against the actual production of saliva if the results were compared. We have not found in the paper a correlation between the sensation related to saliva reported by the patient and the current production of saliva. The relationship between salivary production and the sensation of salivary production is unknown, so it cannot be said that the sensation is subjective, it means that you can not achieved your main objective.

Thank you for your comment. The salivary flow evaluation was conducted at a facility external to our hospital by qualified personnel experienced in performing this procedure. The data that came to our attention was simply a yes or no based on the cutoffs they established for the diagnosis of hyposialia. The patients included in our study were only those who had a high value that did not allow for a diagnosis of hyposialia or a questionable diagnosis [the standard range for unstimulated saliva flow was set at 0.2 mL/min, while the stimulated saliva flow range was set between 0.5–2 mL/min. Patients with an unstimulated salivary flow of less than 0.2 mL/min and a stimulated flow of less than 0.5 mL/min were diagnosed with hyposalivation]. In this way, not having the results in a quantitative and continuous manner, it was not possible to correlate the data with the patient's perception of xerostomia. Surely, it will be a piece of information to be included in future studies on the subject.

The exclusion criteria exclude organic neurological pathologies related to BMS. But psychological disturbances related to BMS are accepted, as well as hypertension or thyroid diseases and treatments for these diseases or even antineoplastic treatments that are known to be related to BMS symptoms and saliva production. At no time in the selection of the sample was specified that the patients were being treated for BMS. The different treatments are not described whether there was a clearance period or how long they had been on treatment. It must be explained.

Thank you for your comment. We clarified in the inclusion criteria that the consecutively enrolled patients were those who were seeking a first-time diagnosis at our facility and were not already undergoing treatment for these symptoms. We have now specified this in our inclusion criteria. Patients who were undergoing treatment with antineoplastic agents were not included, while for other drugs in which literature suggested an association with the BMS symptoms, we performed the challenge-dechallenge-rechallenge test according to our reference (Begaud B. Standardized assessment of adverse drug reactions: the method used in France. Special workshop--clinical. Drug Inf J. 1984;18(3-4):275-81. doi: 10.1177/009286158401800314. PMID: 10268556). This method involves a three-stage process: evaluating three chronological criteria (challenge, dechallenge, and rechallenge); assessing clinical and biological findings; and combining chronological and symptomatic assessments to obtain a 3-degree global score (1: doubtful, 2: possible, 3: probable).

After conducting the test, we excluded 4 patients and we included only those patients whose symptoms remained unchanged throughout the entire procedure and whose underlying condition, such as thyroid abnormalities, previously diagnosed, were currently in a euthyroid state. We clarified this in our methods section.

In the conclusions, the improvement of xerostomia is related to topical treatment, but it is not related to the production of saliva again, so it cannot be concluded if it is about xerostomia or hyposialia.

Thank you for your comment. We have clarified in our conclusions that before issuing a diagnosis or initiating therapy, a thorough salivary and/or instrumental diagnostic evaluation is necessary.

Besides these problems it has not been specified why non-parametrics tests are used for the evaluation of statistical significance.

Thank you for your comment. We have added this information in the methods section specifying that the data were not normally distributed.

Based on these problems of sample selection and evaluation of results, the conclusions reached cannot be considered entirely valid. We believe that these aspects should be changed or at least clarified in order to accept the job.

Thank you for your comment. We have expanded the limitations section and modified the conclusions to specify how sample variability may have influenced the obtained results.

Reviewer 2 Report

The authors present a relevant article on the salivary clinical expression that accompanies burning mouth syndrome. This fact is relevant, since it helps to understand the complexity of the symptomatic expression of this pathology.

We believe that there are some methodological aspects that should be explained more clearly. Authors must clearly define the type of study and whether it is a cross-sectional or longitudinal study. Results that cannot be obtained in a cross-sectional study are shown, such as the therapeutic response to drug treatment with clonazepam.

We think that the causes and characteristics of the excluded patients should be included as complementary material, this would help to better understand the possible comorbidities of patients with BMS.

The data/findings obtained in the imaging tests performed (ultrasound and magnetic resonance imaging) should be included. In addition, it would be interesting to know the results of the sialometry and if there is a correlation with the symptoms described in the different groups.

Another aspect that we believe should be specifically explained is the high results on the HAM-D and HAM-A scales and the absence of any diagnosis of anxiety or depression in the included cases.

Author Response

We believe that there are some methodological aspects that should be explained more clearly. Authors must clearly define the type of study and whether it is a cross-sectional or longitudinal study. Results that cannot be obtained in a cross-sectional study are shown, such as the therapeutic response to drug treatment with clonazepam.

Thank you for your comment. We agree with your comment and apologize for not clarifying this point. We are referring to a cross-sectional study, and the data regarding topical therapy with Clonazepam was only added descriptively because the recruitment lasted more than three months, and the prescription was given only after enrolling the patients and collecting all the anamnestic data and questionnaires. That's why the data related to the therapeutic response to the topical medication was not included in the regression analysis, and if you evaluate this data as inappropriate, we can remove it both from the tables (results) and the discussion sections. The reason for including it was only because it could be useful in discussing the neuropathic nature of salivary symptoms. However, we have clarified the type of study in the Methods section.

We think that the causes and characteristics of the excluded patients should be included as complementary material, this would help to better understand the possible comorbidities of patients with BMS.

Thank you for your comment. We have added an additional file in which we have clarified the reason for the exclusion of the 49 patients who were not enrolled because they did not meet the inclusion and exclusion criteria of the study.

The data/findings obtained in the imaging tests performed (ultrasound and magnetic resonance imaging) should be included. In addition, it would be interesting to know the results of the sialometry and if there is a correlation with the symptoms described in the different groups.

Thank you for your comment. The ultrasound and magnetic resonance imaging tests were performed at an external private facility, personally chosen by the patient. We only used the report data during the patient recruitment phase to include only those patients who did not show alterations that would fall under the study's exclusion criteria. Similarly, the salivary flow evaluation was also conducted at a facility external to our hospital by qualified personnel experienced in performing this procedure. The data that came to our attention was simply a yes or no based on the cutoffs they established for the diagnosis of hyposialia. The patients included in our study were only those who had a high value that did not allow for a diagnosis of hyposialia or a questionable diagnosis [The standard range for unstimulated saliva flow was set at 0.2 mL/min, while the stimulated saliva flow range was set between 0.5–2 mL/min. Patients with an unstimulated salivary flow of less than 0.2 mL/min and a stimulated flow of less than 0.5 mL/min were diagnosed with hyposalivation]. In this way, not having the results in a quantitative and continuous manner, it was not possible to correlate the data with the patient's perception of xerostomia. Surely, it will be a piece of information to be included in future studies on the subject.

Another aspect that we believe should be specifically explained is the high results on the HAM-D and HAM-A scales and the absence of any diagnosis of anxiety or depression in the included cases.

Thank you for your comment. The patients had never been assessed for anxiety and depression before coming under our observation. At time zero, the HAM-D and HAM-A tests were conducted. Subsequently, the patients underwent a psychiatric evaluation (by a GP operator) which confirmed the obtained data, but such information could be subject to study for future works.

Reviewer 3 Report

The work is excellent. Although, the results do not reveal great changes over what "many" of us thought... thanks to this work, it has a statistical support, with high strength. 

Author Response

Thank you for your comment. We have made a few modifications to the work to clarify certain methodological aspects in agreement with the other referees. We have attached the finalized manuscript.

Round 2

Reviewer 1 Report

Dear colleagues,

Article Text has improved in its explanations.

Study design and methods to relate signs of salivary production to symptoms of salivary production sensation should be improved in future research.

Points that needed clarification have been clarified and the text has improved in its explanation and conclusions.

Author Response

Dear reviewer,

thank you for carefully reviewing our manuscript. We have modified the manuscript according to the Academic Editor suggestions. 
